# Formation Thermodynamics of Carbamazepine with Benzamide, Para-Hydroxybenzamide and Isonicotinamide Cocrystals: Experimental and Theoretical Study

**DOI:** 10.3390/pharmaceutics14091881

**Published:** 2022-09-06

**Authors:** Alex N. Manin, Denis E. Boycov, Olga R. Simonova, Tatyana V. Volkova, Andrei V. Churakov, German L. Perlovich

**Affiliations:** 1G. A. Krestov Institute of Solution Chemistry, The Russian Academy of Sciences, 1 Akademicheskaya St., Ivanovo 153045, Russia; 2Institute of General and Inorganic Chemistry, Russian Academy of Sciences, 31 Leninsky Prospekt, Moscow 119991, Russia

**Keywords:** cocrystal, solubility, crystal structure, H-bond propensity, competitive reaction method, cocrystal formation thermodynamics

## Abstract

Formation thermodynamic parameters for three cocrystals of carbamazepine (CBZ) with structurally related coformers (benzamide (BZA), para-hydroxybenzamide (4-OH-BZA) and isonicotinamide (INAM)) were determined by experimental (cocrystal solubility and competitive reaction methods) and computational techniques. The experimental solubility values of cocrystal components at eutectic points and solubility product of cocrystals [CBZ + BZA], [CBZ + 4-OH-BZA], and [CBZ + INAM] in acetonitrile at 293.15 K, 298.15 K, 303.15 K, 308.15 K, and 313.15 K were measured. All the thermodynamic functions (Gibbs free energy, enthalpy, and entropy) of cocrystals formation were evaluated from the experimental data. The crystal structure of [CBZ + BZA] (1:1) cocrystal was solved and analyzed by the single crystal X-ray diffractometry. A correlation between the solubility products and pure coformers solubility values has been found for CBZ cocrystals. The relationship between the entropy term and the molecular volume of the cocrystal formation has been revealed. The effectiveness of the estimation of the cocrystal formation thermodynamic parameters, based on the knowledge of the melting temperatures of active pharmaceutical ingredients, coformers, cocrystals, as well as the sublimation Gibbs energies and enthalpies of the individual components, was proven. A new method for the comparative assessment of the cocrystal stability based on the H-bond propensity analysis was proposed. The experimental and theoretical results on the thermodynamic parameters of the cocrystal formation were shown to be in good agreement. According to the thermodynamic stability, the studied cocrystals can be arranged in the following order: [CBZ + 4-OH-BZA] > [CBZ + BZA] > [CBZ + INAM].

## 1. Introduction

Solubility, stability, hygroscopicity, and melting point are significant parameters affecting the therapeutic efficacy of the drug. The preparation of two-component crystals (cocrystals) makes it possible to obtain the desired physicochemical and biopharmaceutical properties of active pharmaceutical ingredients [1]. The solubility advantage via the formation of cocrystals for poor soluble drugs is widely highlighted throughout the literature [2,3]. At the same time, the stability issue is not less important for the design of the pharmaceutical formulations based on the cocrystals. A cocrystal strategy includes the selection of an appropriate cocrystal former (coformer) which is able not only to increase the API solubility and dissolution rate, but also to improve the stability [4]. In most cases, the coformer screening is a continuous process requiring materials expenditure. From this, an issue of predicting the appropriate coformer to a certain API to the aim of solubility and stability advantage arises. A real path to the prediction of the stability of a two-component crystal composed of two original compounds is the evaluation of the formation Gibbs energy ($\Delta G_{f}^{}$) [5]. To this end, the cocrystal solubility method described by Schartman [6] seems to be promising, among others, and it is able to provide us with exhaustive information on the thermodynamic stability.

The object of the present study is the determination of the thermodynamic stability of three carbamazepine (CBZ) cocrystals. Carbamazepine belongs to the group of widely used antiepileptic drugs. It acts as a stabilizer of the inactivated state of sodium channels resulting in diminishing the excitability of the brain cells [7]. Besides, Wang et al. [8] reported CBZ, as a non-antibiotic pharmaceutical, to be effective in promotion of conjugative transfer of antibiotic resistance genes carried by plasmid within and across bacterial genera and serving as a model gene transfer system for environmental ecosystems. The great potential of using carbamazepine supports further outlining the directions for the production of its promising and stable formulations based on the two-component crystals.

In spite of the fact that there is a vast array of literature sources on the topic of the design of carbamazepine two-component crystals with different coformers [9,10,11], only several works have been identified within the literature on the details of the thermodynamic stability of the CBZ cocrystals [6,12,13,14]. While only two studies [13,14] represent the experiments at different temperatures with the full evaluation of the thermodynamic parameters for carbamazepine–nicotinamide cocrystal in ethanol–water mixtures and carbamazepine–saccharin cocrystal in methanol. Shayanfar et al. [14] used the solubility shake-flask method, whereas the solution calorimetry was applied in the study of Oliveira et al. [13] to the aim of the formation enthalpy determination.

Taking into account the importance of the prediction of the cocrystals stability from the thermodynamic point, in the present study, we determine the thermodynamic formation parameters for three cocrystals: carbamazepine–benzamide [CBZ + BZA], carbamazepine–4-hydroxybenzamide [CBZ + 4-OH-BZA], and carbamazepine–isonicotinamide [CBZ + INAM] in acetonitrile by the cocrystal solubility method (CCSM) and compared the obtained results with the prognostic models reported by Perlovich [5].

The research objects were selected based on the following considerations. First of all, the structurally related compounds were chosen as the coformers to simplify the analysis of the crystal structure effect on the thermodynamic characteristics of the cocrystal formation. Secondly, we dealt with the screening of the cocrystals with the identified crystal structures (either using SCXR or PXRD) which have been already described in the literature. Based on these premises, we wanted to focus our efforts on the solubility study for a detailed investigation of the dissolution processes of the cocrystals and individual compounds thereof at different temperatures. Just such a design of the experiment makes it possible to calculate all the thermodynamic functions of the cocrystal formation and thoroughly analyze the main driving forces of the process. Thirdly, since CBZ is a model compound and active pharmaceutical ingredient of quite a lot of the existing cocrystals with the solved crystal structures, this work is the beginning of a series of the studies on the thermodynamic aspects of the formation of CBZ cocrystals. Our outlook for the future includes the disclosing of the fundamental regularities of cocrystallization not only for this class of compounds, but a generalization of this approach for the two-component molecular crystals on the basis of the vast array of the obtained experimental data.

## 2. Materials and Methods

### 2.1. Materials

The objects of the study: benzamide (BZA, C_7_H_7_NO, 99%), 4-hydroxybenzamide (4-OH-BZA, C_7_H_7_NO_2_, 98%), were obtained from Sigma–Aldrich. Carbamazepine (CBZ, C_15_H_12_N_2_O, 98%) and isonicotinamide (INAM, C_6_H_6_N_2_O, 99%) were purchased from Acros Organics. 

The solvent, acetonitrile (ACN), was from RCI Labscan and used as received without further purification. All solvents used for the crystallization experiments were of analytical grade.

### 2.2. Preparation of Multi-Component Crystals

#### 2.2.1. Slurry Experiment

The multi-component crystals of CBZ: [CBZ + BZA], [CBZ + 4-OH-BZA], and [CBZ + INAM] were prepared by the slurry method. The required amounts of CBZ and conformer were accurately weighted to obtain 1:1 stoichiometry. The needed volume of acetonitrile (ACN) (at approx. 1 mL) was added to the mixture to obtain a suspension which was stirred in a weighing bottle with a magnetic stirrer overnight at room temperature. The obtained cocrystal samples were dried and characterized by PXRD to make sure that the pure cocrystal had been formed.

#### 2.2.2. Solution Crystallization

A slow evaporation method was used to grow single crystals. Cocrystallization experiments were carried out by dissolving 50 mg of a 1:1 mixture of CBZ and BZA in a minimum amount of the acetonitrile (ACN). The vials containing the dissolved mixtures were covered with a parafilm, punctured with 3–5 holes and allowed to slowly evaporate under ambient conditions. Small colourless needles of the 1:1 cocrystal formed after 3 days.

### 2.3. Solid State Characterizations of CBZ Multi-Component Crystals

The powder X-ray diffraction (PXRD) technique was used in the present study for the solid sample characterization. PXRD patterns of the bulk materials were recorded under ambient conditions with the help of a D2 PHASER XRD diffractometer (Bruker, Germany) operating at 30 kV and 10 mA using Cu-K_α1_ radiation (λ = 1.54187 Å). The data were collected in the range of 5–30° 2 theta with a 0.03° step.

### 2.4. Single Crystal X-ray Diffraction (SCXRD)

The X-ray diffraction data were collected on a Bruker SMART APEX II diffractometer with graphite-monochromated Mo-K_α_ radiation (λ = 0.71073 Å). Adsorption corrections based on measurements of equivalent reflections was applied [15]. The structure was solved by direct methods and refined by full matrix least-squares on F^2^ with anisotropic thermal parameters for all the non-hydrogen atoms [16]. All hydrogen atoms were found from difference Fourier maps and refined isotropically. The crystallographic data were deposited with the Cambridge Crystallographic Data Centre (CCDC), CCDC No. 2196018. This information can be obtained free of charge from the Cambridge Crystallographic Data Centre via www.ccdc.cam.ac.uk/data_request/cif (accessed on 10 August 2022).

### 2.5. Differential Scanning Calorimetry (DSC)

The thermal analysis was carried out using a differential scanning calorimeter with a refrigerated cooling system (Perkin Elmer DSC 4000, Waltham, MA, USA). The sample was heated in a sealed aluminum sample holder at a rate of 10 °C·min^−1^ in a nitrogen atmosphere. The unit was calibrated with indium and zinc standards. The accuracy of the weighing procedure was ±0.01 mg.

### 2.6. Competitive Grinding of Three Component Samples

The liquid assisted grinding with acetonitrile as a solvent was applied to qualitatively evaluate the relative stability of the multi-component crystals obtained herein. The physical mixtures of CBZ and two coformers (in different combinations) were ground with the addition of a drop of acetonitrile. A Fritsch planetary micromill, model Pulverisette 7, with 12 mL agate jars and ten 5 mm agate balls at a rate of 500 rpm for 60 min was used for grinding. The result powder products were characterized by PXRD.

### 2.7. Solubility Determination of the Individual Components and Cocrystals

The solubility of individual API (CBZ) and coformers (BZA, 4-OH-BZA and INAM) as well as their multi-component crystals: [CBZ + BZA], [CBZ + 4-OH-BZA], and [CBZ + INAM] in acetonitrile was determined by suspending an excess amount of the material in 2 mL of the solvent at constant shaking for 24 h at 293.15 K, 298.15 K, 303.15 K, 308.15 K and 313.15 K up to the equilibrium. Then, the solution was kept stationary during several hours and filtered through a 0.2-μm syringe filter. The concentration of the compounds was determined by the HPLC method. The solid phases of individual compounds and two-component crystals at equilibrium were characterized by PXRD.

### 2.8. High-Performance Liquid Chromatography (HPLC)

The HPLC was performed with Shimadzu Prominence model LC-20 AD equipped with a PDA detector and a C-18 column Luna^®^ (150 mm × 4.6 mm i.d., 5 μm particle size and 100 Å pore size). The column temperature was set at 40 °C. Elution of the samples was per-formed by a mobile phase consisting of acetonitrile and water mixed in a ratio of 40:60 *v/v* in an isocratic regime using the flow rate of 1 mL·min^−1^. The injection volume was 20 μL. The UV detection of CBZ and coformers (BZA, 4-OH-BZA, INAM) was determined at the wavelengths of 284 nm, 224 nm, 251 nm, and 263 nm, respectively. The retention time of CBZ and coformers (BZA, 4-OH-BZA, INAM) was 4.5 min, 2.1 min, 1.7 min, and 1.4 min, respectively. The examples of chromatograms for each cocrystal components are shown in Appendix A.

### 2.9. Thermodynamic Parameters Calculation

Thermodynamic parameters of the cocrystal formation were determined from the results of the solubility experiments according to the literature [6,17]. To this end, the drug (CBZ) and coformers (BZA, 4-OH-BZA, INAM) solution concentrations in equilibrium with cocrystal and drug phases ${\left[CBZ\right]}_{eu}$ and ${\left[CF\right]}_{eu}$ were measured at 5 particular temperatures (293.15 K, 298.15 K, 303.15 K, 308.15 K and 313.15 K) in order to calculate the cocrystal solubility ($S_{CC}$) for 1:1 cocrystal as follows:(1)$$S_{CC}=\sqrt{{\left[CBZ\right]}_{eu}\cdot {\left[CF\right]}_{eu}}$$
where ${\left[CBZ\right]}_{eu}$ and ${\left[CF\right]}_{eu}$ are the solubility values at equilibrium of drug and conformer, respectively. The term $S_{CC}$ refers to the stoichiometric solubility of the cocrystal, that is, the cocrystal solubility under solution molar ratios equal to those of the cocrystal. Next, the solubility products ($K_{sp}$) of the cocrystals of interest were calculated at particular temperatures by the equation:(2)$$K_{sp}={\left[CBZ\right]}_{eu}{\left[CF\right]}_{eu}={\left(S_{CC}\right)}^{2}$$

The next task was to determine the standard free energy change of the cocrystal formation process ($\Delta G_{f}^{0}$) considering a solution of the drug in equilibrium with the solid phase:(3)$$\Delta G_{f}^{0}=-RT\ln(\frac{S_{}^{0}(CBZ)\cdot S_{}^{0}(CF)}{K_{SP}})=-RT\ln K_{f}$$
where $S_{}^{0}(CBZ)$ and $S_{}^{0}(CF)$ are the solubility values of CBZ and the particular conformer, respectively, in pure form at a particular temperature T, $K_{f}^{}$ is the formation constant. Using the $K_{f}^{}$ values at different temperatures, the standard formation enthalpy ($H_{f}^{0}$) of a multi-component crystal (cocrystal in our case) can be derived from the vant’Hoff relation:(4)$$\frac{d\ln K_{f}}{d(1/T)}=-\frac{\Delta H_{f}^{0}}{R}$$

At last, the formation entropy change can be derived with the help of the general equation knowing the $G_{f}^{0}$ and $H_{f}^{0}$:(5)$$\Delta G_{f}^{0}=\Delta H_{f}^{0}-T\Delta S_{f}^{0}$$

### 2.10. H-Bond Propensity Analysis

The H-bond propensities (HBP) [18,19,20] were calculated using the program Mercury 2021.3.0 [21]. Each pair of CBZ and coformers was sketched and auto-edited, and functional groups were selected as suggested by Mercury. The functional groups defined for the purpose of the H-bond propensity calculation were taken from the library of functional groups provided with the software (the CSD Version 5.43, (updated on Mar 2022)). The MultiComponent Score value (MC score), i.e., the difference between the propensity of the best hetero-interaction and the best homo-interaction, was applied to estimate the probability of cocrystal formation in HBP calculation method [22]. The integrated MC score values for the integrated HBP calculation method (intHBP) were calculated using the following equation:(6)$$MC_{\mathrm{int}}=\sum_{i=1}^{n}MC_{i}(A)+\sum_{j=1}^{m}MC_{j}(B)$$
where $MC_{\mathrm{int}}$ denotes integrated MultiComponent Score; *MC(A)* is the difference between the highest A:B hydrogen bond propensity value and the highest A:A hydrogen bond propensity value for each donor of target molecule; *MC(B)* the difference between the highest B:A hydrogen bond propensity value and the highest B:B hydrogen bond propensity value for each donor of coformer, *n* and m the number of donor in the target molecule and in the coformer, respectively, and *i* and *j* the number of functional group with hydrogen bond donors in the target molecule and in the coformer, respectively.

Using the HBP method to estimate the probability of the existence of polymorphic forms [23,24], we proposed a new approach to assessing the efficiency of the cocrystal packing. This approach consists in the fact that if all the most favorable hydrogen bonds are realized (with the maximum calculated value of the H-bond propensity) during the formation of a cocrystal, then the most “effective” packing of the components is realized and may indicate the greatest stability of the resulting cocrystal form. Since the formation Gibbs energy is a quantitative indicator of the stability of a cocrystal, it was interesting to evaluate how the parameter of the influence of the “efficiency” of the molecules packing in a cocrystal correlates with the value of the formation Gibbs energy.

Packing efficiency of the molecules in a crystal lattice ($\varepsilon_{HBP}$) can be estimated by comparing the sums of the probabilities of the formation of predicted ($\Sigma HBP_{pred}$) and not predicted ($\Sigma HBP_{non-pred}$) hydrogen bonds by the HBP method:(7)$$\varepsilon_{HBP}=\Sigma HBP_{pred}-\Sigma HBP_{non-pred}$$

## 3. Results and Discussion

### 3.1. Crystal Structure Analysis

As mentioned above, knowledge of the crystal structure is an important tool for the interpretation of the experimental data on the thermodynamics of cocrystal formation. First of all, we want to emphasize per se the procedure used for the preparation of the pure cocrystals, which, as a rule, is associated with the comparison of the diffraction patterns of the pure product and the products obtained as a result of the preparation procedure. The diffraction patterns designed from the decoded single crystals of the cocrystals are considered as an ideal case. The literature analysis showed that for the selected cocrystals, the crystal structures from single crystals were decoded for [CBZ + 4-OH-BZA] [25] and [CBZ + INAM] [26]. In turn, for [CBZ + BZA], the crystal structure was described only using PXRD [27]. Taking into account our future plans to study experimentally the thermodynamic stability of the selected cocrystals relatively to each other and bearing in mind that the main criterion for this analysis is the comparison of the crystal diffraction patterns, it becomes relevant to decipher the crystal structure from a single crystal for the [CBZ + BZA] cocrystal. For these purposes, the appropriate conditions (described in the previous section) were chosen, a single crystal of the cocrystal was grown, and the structure was deciphered. The crystallographic data for the [CBZ + BZA] (1:1) cocrystal are given in Table 1.

The colorless needle-shape crystals grown from acetonitrile were found to belong to the monoclinic, *P2_1_/n* space group. The asymmetric unit consists of one carbamazepine and one benzamide molecules (Figure 1a). The arrangement of molecules in the crystal lattices of [CBZ + BZA] (1:1) and [CBZ + INAM] (1:1) form II (CSD refcode LOFKIB01) [26], as well as the identity of their simulated PXRD patterns (Appendix A), prove that the cocrystals are isostructural.

The crystal structure of [CBZ + BZA] (1:1) forms a 4-membered supramolecular unit consisting of two carbamazepine and two benzamide molecules. The molecules are held together by two type of N-H···O hydrogen bonds (N1-H1···O2 (2.05(2) Å, 153(2)°) and N12-H12···O1 (2.30(2) Å, 136(2)°) between the carboxamide group of CBZ and the amide group of BZA which form ring motif with $R_{4}^{4}(16)$ graph set notation (Figure 1b). The four-membered supramolecular units then interact with each other via the N-H···O (2.07(2) Å, 160(2)°) hydrogen bond among BZA molecules to form a chain along the a-axis with a C(4) graph set notation (Figure 1c). The CBZ molecules of the neighbor units form carboxamide- carboxamide homosynthon ($R_{2}^{2}(8)$ ring motif) (Figure 1c,d). The neighboring chains of four-membered supramolecular units interact with each other by the C-H···O, C-H···N contacts and other weak non-covalent interactions (Figure 1d).

### 3.2. Solubility Experiments

The experimental values of the solubility, solubility product ($K_{sp}$) and formation free energy ($\Delta G_{f}^{0}$) of CBZ, coformers and cocrystals [CBZ + BZA], [CBZ + 4-OH-BZA], and [CBZ + INAM] in acetonitrile at 293.15 K, 298.15 K, 303.15 K, 308.15 K and 313.15 K are shown in Appendix A. To prove the isocongruent dissolution of the studied cocrystals in acetonitrile, we checked the diffraction patterns of the bottom phases after the solubility experiments and compared them with those of the individual compounds included in the cocrystals and the cocrystal (Appendix A). The thermodynamic functions of the cocrystals formation at 298 K are summarized in Table 2. In addition, for clarity, Figure 2 shows the temperature dependences for $\ln K_{f}$ (correlation equations for which are given at the footnote of Table 2).

To confirm the exclusive meaningfulness of the entropy contribution to the formation of the studied cocrystals, we tried to estimate the molecular volume of cocrystal formation ($\Delta V_{f}(CC)$), based on the results from the X-ray diffraction data from the single crystals of the cocrystals and individual compounds, using the following equation:(8)$$\Delta V_{f}(CC)=V_{mol}(CC)-(X_{1}\cdot V_{mol}(API)+X_{2}\cdot V_{mol}(CF))$$
where (API)*_n_*(CF)_m_: $X_{1}=n/(n+m)$, $X_{2}=m/(n+m)$, $V_{mol}(CC)=V_{cell}(CC)/Z$, $V_{mol}(API)=V_{cell}(API)/Z$, $V_{mol}(CF)=V_{cell}(CF)/Z$, $V_{cell}$ is the volume of unit cell of cocrystal, API and CF respectively, Z is a number of molecules in the unit cell.

Since five polymorphic modifications have been described for carbamazepine in the literature [28] the most thermodynamically stable polymorph was chosen for calculations (Form III, P21/c). The calculated values of $\Delta V_{f}(CC)$ are given in Table 2. Evidently, for [CBZ + 4-OH-BZA], $\Delta V_{f}(CC)$ is negative (−19.0 Å^3^). In other words, during the formation of a given cocrystal, the molecular volume in the cocrystal is less than that of the corresponding mixture of the molecular volumes of individual compounds. Moreover, this indicates an increase in the ordering in the cocrystal and, as a consequence, a decrease in the entropy. In contrast, the positive $\Delta V_{f}(CC)$ values (6.2 and 6.1 Å^3^, respectively) are observed for the [CBZ + BZA] and [CBZ + INAM] cocrystals, which indicates an increase in the entropy in the cocrystal as compared to the crystals of the individual compounds. The obtained values are in good agreement with the entropy terms of the cocrystals formation $T\Delta S_{f}^{298}(\exp)$: for [CBZ + 4-OH-BZA] corresponds to −10.7, for [CBZ + BZA] corresponds to 30.5 and for [CBZ + INAM] corresponds to 22.0 kJ·mol^−1^. Thus, the cocrystallization reaction for [CBZ + 4-OH-BZA] will be the most favorable under the temperature decrease. However, when obtaining a cocrystal, in addition to the thermodynamic factor, the kinetic factor (minimum activation barriers for the processes of nucleation and growth) should be taken into account. Unfortunately, only one [CBZ + Saccharin] (1:1) cocrystal, for which all the thermodynamic functions of cocrystal formation obtained by the isothermal saturation method in isoeutectic points at 33 °C in methanol were calculated, is described in the literature [13]. Notably, in this work (similarly to our study) Form III of CBZ was selected for calculations (Table 2). It is evident that in terms of the behavior of the formation thermodynamic functions and by the $\Delta V_{f}(CC)$ value, the cocrystal with Saccharin is very similar to the cocrystal with 4-OH-BZA. In addition, it should be noted that for the cocrystals considered in Table 2, there is a trend between $T\Delta S_{f}^{}(\exp)$ and $\Delta V_{f}(CC)$ (Appendix A). Unfortunately, the number of points does not allow us to obtain correlation dependence, but in the future, we plan to expand the number of objects under study.

Predicting the solubility of a cocrystal based on the only simple physicochemical characteristics of the individual compounds included in it is a very important and intriguing problem. The attempts for such a description have been made in the literature, and among them we would like to note the correlation models connecting the solubilities of the individual compounds and their physicochemical descriptors with the discussed value [29,30,31]. For example, for these purposes Avdeef [29] used the Abraham descriptors, whereas, in our studies we applied the HYBOT descriptors [32]. In the present work, we tried to analyze the solubilities of only CBZ-based cocrystals. Therefore, we reduced the number of descriptors to a minimum (all the more, the number of the experimental values on the solubility of CBZ cocrystals described in the literature does not allow to construct serious correlation models). The solubility of the individual coformers $S^{0}(CF)$ was considered as a descriptor for predicting the $K_{sp}$ values of the cocrystals. Appendix A contains the literature data used for the correlations. The results of the correlation analysis are shown in Figure 3.
Figure 3Plot of log(Ksp) versus log(S0(CF)). Numbering corresponds to Appendix A and can be described by the following equation:
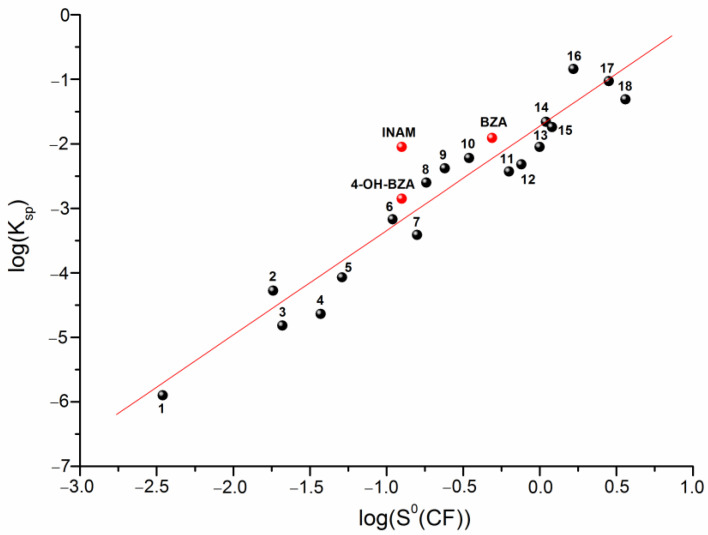

(9)$$\log(K_{sp})=(-1.72\pm 0.11)+(1.62\pm 0.12)\cdot \log(S^{0}(CF))\;R=0.9488;\;\mathrm{SD}=0.43;\;n=21$$

Here, we would like to note that any solvents for which the equilibrium values of the individual components of the cocrystal at eutectic points can be measured (in the presence of an equilibrium bottom phase) can be used for consideration. Thus, we have a rather simple correlation equation, making it possible to predict the solubility of a CBZ cocrystal based only on the solubility of the pure coformer in the solvent in which the solubility of the cocrystal is supposed to be predicted.

### 3.3. Cocrystals Formation Thermodynamics

The approach to prediction of thermodynamic functions of cocrystal formation ($\Delta G_{f}^{0,298}(CC)$, $\Delta H_{f}^{0,298}(CC)$) based on the knowledge of melting temperatures of active pharmaceutical ingredients (API) ($T_{m}(API)$), coformers (CF) ($T_{m}(CF)$), cocrystals ($T_{m}(CC)$), as well as the sublimation Gibbs energies and enthalpies of individual components included in the cocrystal ($\Delta G_{sub}^{0,298}(API)$, $\Delta G_{sub}^{0,298}(CF)$, $\Delta H_{sub}^{0,298}(API)$, $\Delta H_{sub}^{0,298}(CF)$) was previously developed by us [5].

The literature data show that in structurally similar compounds (according to Tanimoto similarity coefficients) there is a linear correlation between $\Delta G_{sub}^{0,298}$ and $T_{fus}$ [31]. The crystal structure of a two-component co-crystal (with any stoichiometry) can be considered structurally similar to that of individual compounds. In other words, the experimental values of $\Delta G_{sub}^{0,298}$ vs. $T_{fus}$ of the individual compounds and the co-crystal belong to one (and the same) cluster. Therefore, coefficients A and B of Equation (10) can be calculated if $\Delta G_{sub}^{0,298}$ and $T_{fus}$ of the individual compounds are known.
(10)$$\Delta G_{sub}^{0,298}=A+B\cdot T_{fus}$$

Knowing coefficients A, B, and $T_{fus}(CC)$, we can calculate the $\Delta G_{sub}^{0,298}(CC)$-value:(11)$$\Delta G_{sub}^{0,298}(CC)=A+B\cdot T_{fus}(CC)$$

In order to calculate the Gibbs energy of co-crystal formation with the stoichiometric ratio (API)*_n_*(CF)_m_ $\Delta G_{f}^{0,298}(CC)$, we should use the following equations:(12)$$\Delta G_{sub}^{0,298}(PM)=X_{1}\cdot \Delta G_{sub}^{0,298}(API)+X_{2}\cdot \Delta G_{sub}^{0,298}(CF)$$
where (API)*_n_*(CF)_m_: X_1_ = *n*/(*n* + m); X_2_ = m/(*n* + m). $\Delta G_{sub}^{0,298}(PM)$, $\Delta G_{sub}^{0,298}(API)$, $\Delta G_{sub}^{0,298}(CF)$ are the sublimation Gibbs energies of the physical mixture, API and CF, respectively.
(13)$$\Delta G_{f}^{0,298}(CC)=\Delta G_{sub}^{0,298}(PM)-\Delta G_{sub}^{0,298}(CC)$$

The co-crystal formation enthalpy value, $\Delta H_{f}^{0,298}(CC)$, was obtained by the following algorithm. It is well known that there is a linear dependence between $\Delta G_{sub}^{0,298}$ and $\Delta H_{sub}^{0,298}$ (the so-called compensation effect) [33]. Thus, knowing the experimental values $\Delta G_{sub}^{0,298}$ and $\Delta H_{sub}^{0,298}$, it is possible to calculate the coefficients of Equation (14):(14)$$\Delta G_{sub}^{0,298}=C+D\cdot \Delta H_{sub}^{0,298}$$

Then, calculate the necessary value:(15)$$\Delta H_{f}^{0,298}(CC)=\Delta H_{sub}^{0,298}(PM)-\Delta H_{sub}^{0,298}(CC)$$
(16)$$\Delta H_{sub}^{0,298}(CC)=(\Delta G_{sub}^{0,298}(CC)-C)/D$$
(17)$$\Delta H_{sub}^{0,298}(PM)=X_{1}\cdot \Delta H_{sub}^{0,298}(API)+X_{2}\cdot \Delta H_{sub}^{0,298}(CF)$$

Since all the necessary parameters for the studied cocrystals are known, it is possible to evaluate the formation thermodynamic functions and to compare them with the experimental values. Table 2 represents the calculation results. In turn, all the data on the sublimation thermodynamics and melting points of the individual compounds are given in Appendix A. The diagrammatic approach was used in order to analyze the experimental and theoretically obtained values of the thermodynamic functions of the cocrystals formation. The results of the analysis are presented in Figure 4 as a diagram, where the abscissa corresponds to the formation enthalpy ($\Delta H_{f}^{0,298}$), and the ordinate corresponds to the entropic term ($T\Delta S_{f}^{0,298}$). The dotted lines correspond to the isoenergetic cocrystal formation Gibbs energy values ($\Delta G_{f}^{0,298}$). The diagram is divided into eight sectors, each corresponding to a different ratio of the enthalpy and entropy contributions to the Gibbs energy. Each sector is formed by two lines: on the one side, the line corresponding to the zero $\Delta H_{f}^{0,298}$ or $T\Delta S_{f}^{0,298}$ value; on the other side, the bisector of the angles formed at the intersection of the coordinates ($\Delta H_{f}^{0,298}$;$T\Delta S_{f}^{0,298}$). Thus, the diagram can be divided into the following areas: ($T\Delta S_{f}^{0,298}$>$\Delta H_{f}^{0,298}$ > 0) ≡ sector **A**, ($\Delta H_{f}^{0,298}$ < 0; $T\Delta S_{f}^{0,298}$ > 0; |$T\Delta S_{f}^{0,298}$| > |$\Delta H_{f}^{0,298}$|) ≡ sector **B**, ($T\Delta S_{f}^{0,298}$<$\Delta H_{f}^{0,298}$ < 0) ≡ sector **E**, and ($\Delta H_{f}^{0,298}$ > 0; $T\Delta S_{f}^{0,298}$ < 0; |$T\Delta S_{f}^{0,298}$| > |$\Delta H_{f}^{0,298}$|) ≡ sector **F** belonging to the entropy determined processes. The segments of the diagram where ($\Delta H_{f}^{0,298}$ < 0; $T\Delta S_{f}^{0,298}$ > 0; |$\Delta H_{f}^{0,298}$| > |$T\Delta S_{f}^{0,298}$|) ≡ sector **C**, ($\Delta H_{f}^{0,298}$ < 0; $T\Delta S_{f}^{0,298}$ < 0; |$\Delta H_{f}^{0,298}$| > |$T\Delta S_{f}^{0,298}$|) ≡ sector **D**, ($\Delta H_{f}^{0,298}$>$T\Delta S_{f}^{0,298}$ > 0) ≡ sector **H** и ($\Delta H_{f}^{0,298}$ > 0; $T\Delta S_{f}^{0,298}$ < 0; |$\Delta H_{f}^{0,298}$| > |$T\Delta S_{f}^{0,298}$|) ≡ sector **G** correspond to the enthalpy determined processes.

The experimental values are marked in blue, while the calculated values are marked in red. First, all the calculated $\Delta G_{f}^{0,298}$ values are less than zero, which means that the algorithm developed by us correctly predicts the probability of the formation of the cocrystals considered in this study, i.e., the cocrystals are thermodynamically stable as compared to the corresponding stoichiometric mixture of the individual compounds. An important point of the validity of the calculation technique is that the respective calculated and experimental values belong to the same sector of the diagram (this indicates the same driving forces of the cocrystal formation process). Evidently, the discussed values for INAM and 4-OH-BZA are in the same sectors: for the first cocrystal-in sector A, for the second one-in sector D. Unfortunately, for BZA, the experimental value is in sector A, while the calculated one is in sector D. If we consider the experimental values of the thermodynamic functions of cocrystal formation, we can conclude that for CBZ cocrystals with INAM and BZA, the analyzed process is entropy determined with positive values of the enthalpy and entropy terms. Apparently, the consequence of this is the positive values of $\Delta V_{f}(CC)$. In turn, for the CBZ cocrystal with 4-OH-BZA, the formation process is enthalpy-determined with negative values for both the enthalpy and entropy terms. Most probably, the negative value of the entropy term results in negative $\Delta V_{f}(CC)$.

### 3.4. H-Bond Propensity Analysis

The results of the H-bond propensity calculation performed on the [CBZ + BZA], [CBZ + 4-OH-BZA] and [CBZ + INAM] systems are given in Appendix A. It can be seen that all possible hydrogen bonds are realized in the [CBZ + BZA] (1:1) crystal structure. The highest value of the H-bond propensity for the cocrystals with benzamide and 4-hydroxybenzamide is observed for N-H···O amide-carboxamide heterosynthon. Interestingly, that for [CBZ + BZA] (1:1) this hydrogen bond is formed, whereas it is absent in the [CBZ + 4-OH-BZA] crystal structure [25]. Markedly, the hydroxyl functional group plays a structure-forming role in the [CBZ + 4-OH-BZA] cocrystal. This group acts as both a donor and an acceptor in the formation of two heterosynthons (interactions 4 and 8). Importantly, interaction 8 has the lowest propensity value among all A:B or B:A possible interactions in the [CBZ + 4-OH-BZA] cocrystal.

It should be noted that in all the studied cocrystals, the majority of donors and acceptors are involved in the formation of two hydrogen bonds. Only in the [CBZ + 4-OHBZA] cocrystal the donor and acceptor of the amide functional group take part in the formation of only one hydrogen bond-the amide-amide homosynthon. Moreover, the nitrogen atom of the pyridine ring in the INAM does not participate in the hydrogen bonding of the [CBZ + INAM] cocrystal [26,34]. The packing of CBZ and INAM molecules in the cocrystal crystal lattice is implemented in such a way that the N_pyridine_ participates only in the C-H···N bonds formation. Interestingly, the HBP calculation shows that this N_pyridine_ atom has one of the highest probabilities (interactions 1 and 3) of participating in the formation of both homo- and heterosynthons in the [CBZ + INAM] cocrystal.

The H-bond calculation method is often used to direct cocrystal screening [35,36], so it seemed to be interesting to evaluate the cocrystal prediction accuracy. The main principle of the hydrogen bond propensity screening method (HBP) is the determination of the difference between the best hetero- and homo- interactions propensity values. A positive value indicates that there is a strong hydrogen bonding-based drive towards a cocrystal, whilst a value close to zero suggests that either outcome is feasible. In one of our previous works, we proposed to modify the HBP screening method. The results of the HBP analysis should be considered from the point of view of the “integrated” value (intHBP) of all hydrogen bonds formation competitive probability. The results of HBP calculations analysis are presented in Table 3.

Based on both HBP and intHBP calculation results, it can be argued that the most prospective cocrystal is [CBZ + 4-OH-BZA] (1:1). The less prospective, with negative results of HBP and intHBP calculation, is [CBZ + INAM] (1:1). This agrees well with the formation Gibbs energy, where the most stable cocrystal is [CBZ + 4-OH-BZA] (1:1) with $\Delta G_{f}^{0}$ = −6.7 kJ·mol^−1^, while the less stable is [CBZ + INAM] (1:1) ($\Delta G_{f}^{0}$ = −2.1 kJ·mol^−1^).

To explain the method for evaluating the efficiency of the molecular packing in a cocrystal, we used the example of [CBZ + 4-OH-BZA] (1:1) cocrystal. The H-bond propensity calculation for the cocrystal showed that for the N26 donor, the formation of a hydrogen bond with the O1 acceptor (propensity value equal to 0.80) is more likely than with the O27 acceptor (0.73). When considering a pair of N18-O1 and N18-O27 (Appendix A), the first pair with propensity value equal to 0.60 is preferred. The formation of an O28-O1 (0.53) hydrogen bond is more likely than O28-O27 (0.44). Thus, the analysis of the table of the hydrogen bond formation probabilities for the [CBZ + 4-OH-BZA] cocrystal indicates that the most “efficient” packing of the molecules in the crystal lattice will be one where the hydrogen bond pattern will consist of the N26-O1, N18-O1 and O28-O1 hydrogen bonds. Then the “efficiency” of the cocrystal packing is: ε_HBP_ = 0.73 + 0.6 + 0.53−0.8 = 1.06.

To calculate the packing efficiency of the molecules, only those donors were considered for which the probabilities of the formation of both homo- and hetero-synthons were calculated. Therefore, the hydrogen bonds N26-O28, N18-O28, O28-O28 for [CBZ + 4-OH-BZA] (1:1) and N27-O26 with N18-O26 for [CBZ + INAM] (1:1) were not considered in the calculations.

The obtained values of the packing efficiency of the studied cocrystals can be arranged as follows: [CBZ + INAM] < [CBZ + BZA] < [CBZ + 4-OH-BZA] which is in good agreement with the absolute values of the formation Gibbs energies derived from the experiment. Thus, the presented analysis of the calculated H-bond propensity makes it possible to predict the relative stability in a series of the structurally related cocrystals.

### 3.5. Competitive Reactions Method

One of the alternative experimental methods for the qualitative assessment of the comparative stability of cocrystals is the method of competitive reactions [37,38]. The main idea of this method is the mixing of the API and two competing coformers. Upon the grinding of a three-component mixture with the addition of a solvent in which the cocrystals of these components are stable, the most thermodynamically stable cocrystal with the largest value of the formation Gibbs energy (in the absolute value) should be formed. In our study, the three-component physical mixtures ((CBZ + 4-OH-BZA + BZA), (CBZ + 4-OH-BZA + INAM) and (CBZ + INAM + BZA)) with the (1:1:1) stoichiometry were processed by the liquid-assisted grinding using acetonitrile as a solvent. The diffraction patterns of the obtained samples in comparison with those of the individual components and calculated multi-component crystals are given in Figure 5, Appendix A. The grinding of CBZ and 4-OH-BZA together with both competing coformers (BZA or INAM) led to the [CBZ + 4-OH-BZA] (1:1) cocrystal formation with traces of the unreacted BZA or INAM. The competitive reaction method confirms that the [CBZ + 4-OH-BZA] (1:1) cocrystal is the most thermodynamically favorable among the studied cocrystals.

When CBZ was ground together with BZA and INAM, the PXRD pattern demonstrates all the peaks characteristic of [CBZ + BZA] (1:1) (Figure 5). However, as we have noted earlier, [CBZ + BZA] (1:1) and [CBZ + INAM] form II are isostructural, and their PXRD patterns are practically identical (Appendix A). In this connection, for the identification of the sample forming as a result of the competitive liquid-assisted grinding of the three-component mixture, one should consider the peaks characteristic for the unreacted ingredients. Since the peaks characterizing both benzamide (8°) and isonicotinamide (18°, 23.5°) are visible on the diffractogram of the ground mixture, we can propose that in the process of the competitive reaction with CBZ, BZA and INAM both the [CBZ + BZA] (1:1) and [CBZ + INAM] form II cocrystals are formed. Thus, according to the results of the competitive reactions, the thermodynamic stability values of the CBZ cocrystals, go down as follows: [CBZ + 4-OH-BZA] > [CBZ + BZA] ≈ [CBZ + INAM], which agrees with the results of the experimental and theoretical evaluations shown above.

## 4. Conclusions

The present work discusses several methods for the determination of the cocrystal formation thermodynamic parameters. The thermodynamic formation parameters for three cocrystals: carbamazepine–benzamide [CBZ + BZA], carbamazepine–4-hydroxybenzamide [CBZ + 4-OH-BZA], and carbamazepine–isonicotinamide [CBZ + INAM] in acetonitrile were determined by the cocrystal solubility method and the prognostic model based on the knowledge of melting temperatures of API, coformers and cocrystals, as well as the sublimation Gibbs energies and enthalpies of the cocrystals components. The correlation equation making it possible to predict the solubility of a CBZ cocrystal based only on the solubility value of the pure coformer was derived based on the analysis of the experimental and literature data on the dissolution parameters of the carbamazepine cocrystals. The experimental results demonstrated that the most stable cocrystal is [CBZ + 4-OH-BZA] with $\Delta G_{f}^{298}(\exp)$ = −6.7 ± 0.5 kJ·mol^−1^. The correspondence between the experimental and calculated results was analyzed using the diagrammatic approach. It was found that a decrease in the value of molecular volume of the cocrystal formation leads to a decrease in the entropy of the formation process. The influence of the specific features of the molecular packing in the crystal lattices on the thermodynamic parameters of the cocrystals formation was discussed using the HBP method. The approach to the estimation of the efficiency of the molecular packing in a crystal structure was proposed. The packing efficiency of the molecules was found to correlate with the Gibbs energy parameter of the cocrystal formation The Gibbs energy increases with an increase in the number of the realized hydrogen bonds, which have the maximum calculated values of the H-bond propensities. The competitive reactions method proved the maximal [CBZ + 4-OH-BZA] cocrystal stability among the investigated cocrystals. The competitive (simultaneous) grinding of CBZ with BZA and INAM led to the [CBZ + BZA] and [CBZ + INAM] form II cocrystals mixture.

## Figures and Tables

**Figure 1 pharmaceutics-14-01881-f001:**
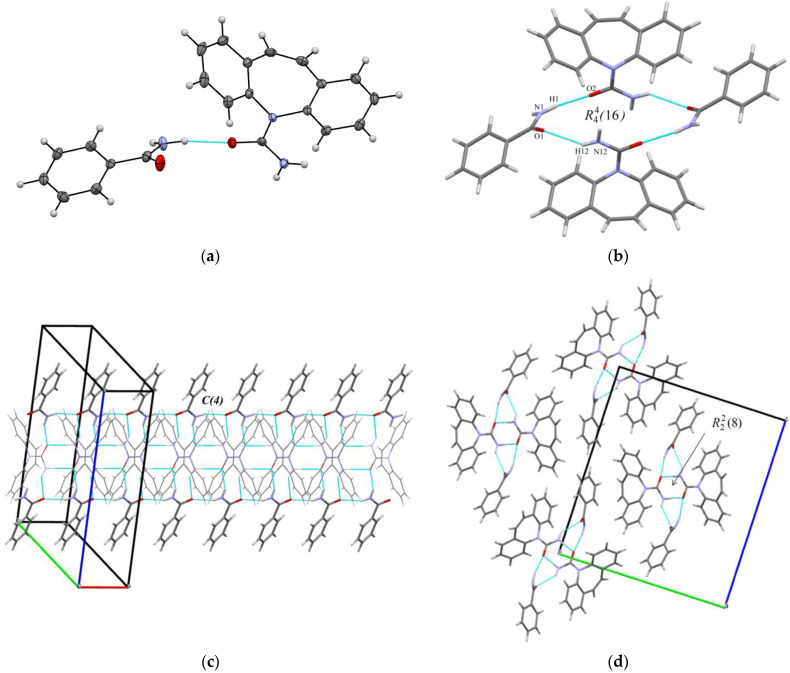
(**a**) Asymmetric unit in [CBZ + BZA] (1:1). The displacement ellipsoids are shown at the 50% probability level. (**b**) Hydrogen bonded packing unit in [CBZ + BZA] (1:1); (**c**,**d**) Molecular packing projections for [CBZ + BZA] (1:1).

**Figure 2 pharmaceutics-14-01881-f002:**
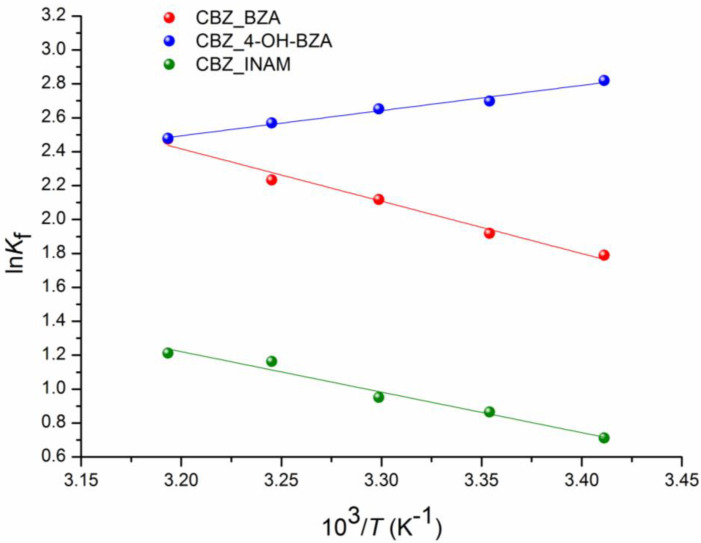
Plots of temperature dependences of lnK_f_ in can: [CBZ + BZA] (1:1) (●), [CBZ + 4-OH-BZA] (1:1) (●), [CBZ + INAM] (1:1) (●).

**Figure 4 pharmaceutics-14-01881-f004:**
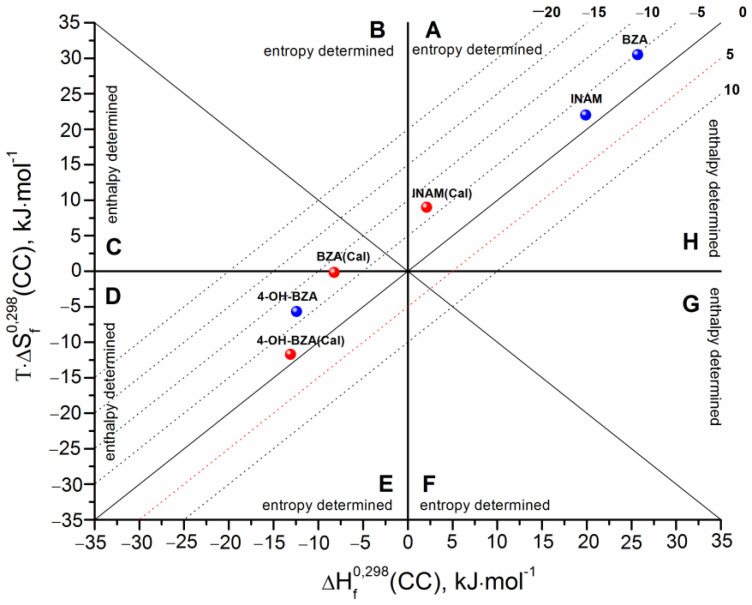
Thermodynamic functions of cocrystal formation processes in coordinates of entropy term vs. enthalpy. The isoenergetic curves of the $\Delta G_{f}^{0,298}(CC)$ function are marked by dotted lines. The red points correspond to calculated values, whereas the blue ones correspond to experimental data.

**Figure 5 pharmaceutics-14-01881-f005:**
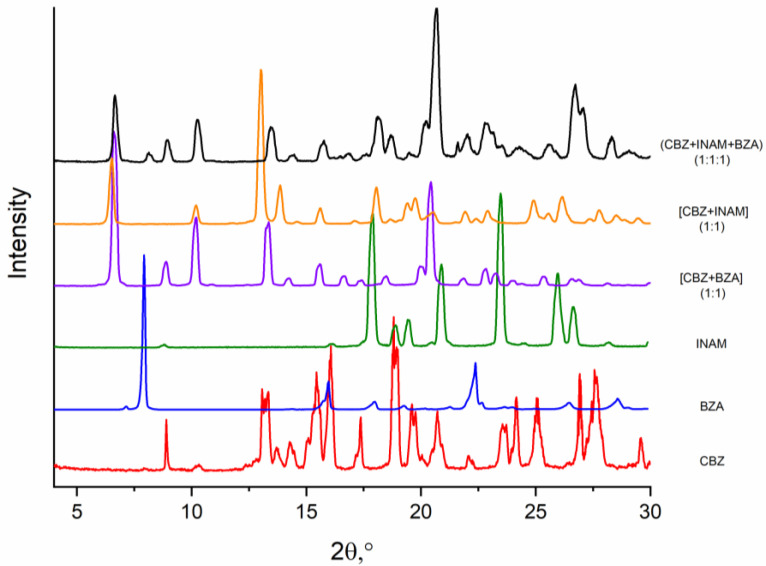
Comparison of the PXRD patterns: CBZ (red), BZA (blue), INAM (green), [CBZ + BZA] (1:1) (violet), [CBZ + INAM] (1:1) (orange) and (CBZ + INAM + BZA) (1:1:1) (black).

**Table 1 pharmaceutics-14-01881-t001:** Crystallographic data and structural refinement summary of the [CBZ + BZA] (1:1).

Chemical Formula	C_15_H_12_N_2_O·C_7_H_7_NO
Formula weight	357.40
Crystal system	Monoclinic
Space group	*P*2_1_/*n*
Crystal size	0.18 × 0.03 × 0.02
Temperature, K	100(2)
*a*, Å	5.0522(2)
*b*, Å	17.6934(8)
*c*, Å	19.9582(7)
*β*, º	90.3848(16)
Volume, Å^3^	1784.04(12)
Calc. density, g·cm^−3^	1.331
*Z*	4
Total reflections	15,579
Independent reflections	3901
*R* _int_	0.0417
*R*_1_ (all data)	0.0662
w*R*_2_ (all data)	0.1043
*R*_1_[*I* > 2*σ*(*I*)]	0.0491
w*R*_2_[*I* > 2*σ*(*I*)]	0.0980
GOF	1.069
CCDC no.	2196018

**Table 2 pharmaceutics-14-01881-t002:** Thermodynamic functions of the cocrystals formation at 298 K evaluated by solubility experiments in acetonitrile and theoretical values calculated using the approach described in [5].

Cocrystal	$\Delta G_{f}^{298}(\exp)[\mathrm{kJ}\cdot {\mathrm{mol}}^{-1}]$	$\Delta H_{f}^{298}(\exp)\;[\mathrm{kJ}\cdot {\mathrm{mol}}^{-1}]$	$T\Delta S_{f}^{298}(\exp)\;[\mathrm{kJ}\cdot {\mathrm{mol}}^{-1}]$	$\Delta G_{f}^{298}(cal)\;[\mathrm{kJ}\cdot {\mathrm{mol}}^{-1}]$	$\Delta H_{f}^{298}(cal)\;[\mathrm{kJ}\cdot {\mathrm{mol}}^{-1}]$	$T\Delta S_{f}^{298}(cal)\;[\mathrm{kJ}\cdot {\mathrm{mol}}^{-1}]$	^ **5** ^ $\Delta V_{f}^{}(CC{)}^{\;}[{Å}^{3}]$
^1^ [CBZ + BZA] (1:1)	−4.8 ± 0.3	25.7 ± 1.9	30.5 ± 2.9	−8.0	−8.2	−0.2	6.2
^2^ [CBZ + 4-OH-BZA] (1:1)	−6.7 ± 0.5	−12.4 ± 0.9	−5.7 ± 0.5	−1.5	−13.1	−11.7	−19.0
^3^ [CBZ + INAM] (1:1)	−2.1 ± 0.1	19.9 ± 1.9	22.0 ± 2.5	−7.0	2.1	9.0	6.1
^4^ [CBZ + Saccharin] (1:1)	−4.6 ± 0.1	−5.9 ± 0.9	−1.3 ± 0.3	0.4	3.5	3.1	−2.5

^1^ lnK_f_ = (12.30 ± 0.74)-(3.09 ± 0.22)·10^3^/T; R = 0.99213; σ = 4.50·10^−3^; *n* = 5; ^2^ lnK_f_ = (−2.26 ± 0.34) + (1.49 ± 0.10)·10^3^/T; R = 0.99283; σ = 9.50·10^−4^; *n* = 5; ^3^ lnK_f_ = (8.86 ± 0.74)-(2.39 ± 0.22)·10^3^/T; R = 0.98721; σ = 4.41·10^−3^; *n* = 5; ^4^ Ref. [13]; ^5^ Calculated by Equation (8).

**Table 3 pharmaceutics-14-01881-t003:** MultiComponent score values, integrated MultiComponent score values and packing efficiency for studied systems.

	[CBZ + BZA] (1:1)	[CBZ + 4-OH-BZA] (1:1)	[CBZ + INAM] (1:1)
MC score (HBP)	0.08	0.07	−0.05
$MC_{\mathrm{int}}$(intHBP)	−0.03	0.08	−0.09
ε_HBP_	0.19	1.06	−0.12

## Data Availability

The results obtained for all experiments performed are shown in the manuscript and Appendix A, the raw data will be provided upon request.

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
