# Peer review of "Formation Thermodynamics of Carbamazepine with Benzamide, Para-Hydroxybenzamide and Isonicotinamide Cocrystals: Experimental and Theoretical Study"

_pharmaceutics, 2022, doi:10.3390/pharmaceutics14091881_

Round 1

Reviewer 1 Report

The paper ‘Formation thermodynamics of Carbamazepine with Benzamide, para-Hydroxybenzamide and Isonicotinamide cocrystals: Experimental and theoretical study’ seems well prepared and worthwhile. The conclusions seem to be supported by the results of the experiments. I recommend it for publication, but I have some comments that should be considered.

1. The ability of carbamazepine to form molecular complexes is well known. There are many reports in the literature about its co-crystals and their properties. The authors presented a rather extensive literature review, in my opinion it is worth mentioning a few more works to show richness of research (for example: 10.1021/cg701022e, 10.1016/j.molstruc.2018.04.100, 10.3390/pharmaceutics9040054, 10.1002/nano.202000201).

2. In sectio 2.4: it seems to me that it is useful to include the CSD structure deposit number..

3. Line 241-243: "The crystal structure of [CBZ + BZA] (1:1) shows 241 the same packing arrangement as that of [CBZ + INAM] (1:1) form II (CSD refcode 242 LOFKIB01) [22]."

Is it to be understood that they are isostructural?

4. The work contains typos and is underdeveloped in terms of editing. This should be corrected but at this stage it did not significantly affect my assessment. 

Author Response

Reviewer_1

The paper ‘Formation thermodynamics of Carbamazepine with Benzamide, para-Hydroxybenzamide and Isonicotinamide cocrystals: Experimental and theoretical study’ seems well prepared and worthwhile. The conclusions seem to be supported by the results of the experiments. I recommend it for publication, but I have some comments that should be considered.

Reviewer_1

  1. The ability of carbamazepine to form molecular complexes is well known. There are many reports in the literature about its co-crystals and their properties. The authors presented a rather extensive literature review, in my opinion it is worth mentioning a few more works to show richness of research (for example: 10.1021/cg701022e, 10.1016/j.molstruc.2018.04.100, 10.3390/pharmaceutics9040054, 10.1002/nano.202000201).

Answer

The suggested works was added to the article.

Reviewer_1

  1. In sectio 2.4: it seems to me that it is useful to include the CSD structure deposit number..

Answer

The CSD structure deposit number was included.

Reviewer_1

  1. Line 241-243: "The crystal structure of [CBZ + BZA] (1:1) shows 241 the same packing arrangement as that of [CBZ + INAM] (1:1) form II (CSD refcode 242 LOFKIB01) [22]."

Is it to be understood that they are isostructural?

Answer:

Yes. It means that [CBZ + BZA] (1:1) and [CBZ + INAM] (1:1) form II are isostructural.

Reviewer_1

  1. The work contains typos and is underdeveloped in terms of editing. This should be corrected but at this stage it did not significantly affect my assessment.

Answer

The text was revised and corrected.

Reviewer 2 Report

This manuscript deals with the cocrystal formation thermodynamic parameters, which were determined by the experimental and computational techniques. All the thermodynamic functions (Gibbs free energy, enthalpy and entropy) of cocrystals formation are evaluated from the experimental data.

  I have the following questions;

1.    I guess that the composition of cocrystal may change with the solution composition.  Did you confirm the relationship between the cocrystal structure and compositions, with the solution compositions in equilibrium through all the experiment?

2.    How do you use the melting points of active pharmaceutical ingredients, conformers, cocrystals for the estimation of cocrystal formation thermodynamic stability parameters. Please explain theoretically.

3.    I think it had better that some important data in Table S2  and in Figure S3 or S4 ( XRD ) are also included in the manuscript .

4.    The effect of activity coefficient for each components can be neglected? It is supposed to be 1.0

5.    As for the relationship between the thermodynamic stability and solution compositions for cocrystal (i.e. clathrate) and the crystallization process of cocrystals, the following references may be usable.

(1)   J.Crystal Growth, 102,255-261(1990)

(2)   Chem. Eng, Res. Des., 86, 1053-1058(2008)

Author Response

Reviewer_2

This manuscript deals with the cocrystal formation thermodynamic parameters, which were determined by the experimental and computational techniques. All the thermodynamic functions (Gibbs free energy, enthalpy and entropy) of cocrystals formation are evaluated from the experimental data.

I have the following questions;

Reviewer_2

  1. I guess that the composition of cocrystal may change with the solution composition.  Did you confirm the relationship between the cocrystal structure and compositions, with the solution compositions in equilibrium through all the experiment?

Answer

Yes, we checked the composition of the cocrystal during the experiment. Supplementary materials contain the results of PXRD analysis of bottom phases. Throughout the experiment, the composition of the solution remained unchanged. The stabilities of the cocrystals in solution are affected by the solvent used. We chose experimentally acetonitrile as a solvent, in which the cocrystal remains stable.

Reviewer_2

  1. How do you use the melting points of active pharmaceutical ingredients, conformers, cocrystals for the estimation of cocrystal formation thermodynamic stability parameters. Please explain theoretically.

Answer

The explanation added to the text of the article (blue color).

Reviewer_2

  1. I think it had better that some important data in Table S2 and in Figure S3 or S4 ( XRD ) are also included in the manuscript .

Answer

Figure S3 has been added to the text of the article. Since the data in Table S2 is not discussed in detail, we believe that it should be left in the SI so as not to bloat an already quite voluminous manuscript.

Reviewer_2

  1. The effect of activity coefficient for each components can be neglected?It is supposed to be 1.0

Answer

Yes it is. The activity coefficient for the solids in equilibrium is equal to 1.

Reviewer_2

  1. As for the relationship between the thermodynamic stability and solution compositions for cocrystal (i.e. clathrate) and the crystallization process of cocrystals, the following references may be usable.

(1)   J.Crystal Growth, 102,255-261(1990)

(2)   Chem. Eng, Res. Des., 86, 1053-1058(2008)

Answer

The compositions of the solutions for each cocrystal remain unchanged (which is proved by the PXRD analysis of the bottom phase and HPLC analysis of the solutions). To study the effect of a solvent on the stability of cocrystals and the crystallization process, it is likely that the articles you have submitted will be extremely useful, but we did not set such a task in the framework of our study.

Reviewer 3 Report

The manuscript presented by Manin et al., discussed the formation thermodynamics of carbamazepine cocrystals. The manuscript is very complete and can be used as a research reference by other scientists. In my opinion the manuscript can be accepted after two small corrections:

1-    Lines 70-75: The authors should clarify the origin of the carbamazepine cocrystals. Where they already described in the literature?

2-    Line 157. Please insert the column brand. Also, add a representative chromatogram of each cocrystal analyzed in Supplementary section.

Author Response

Reviewer_3

The manuscript presented by Manin et al., discussed the formation thermodynamics of carbamazepine cocrystals. The manuscript is very complete and can be used as a research reference by other scientists. In my opinion the manuscript can be accepted after two small corrections:

Reviewer_3

1 - Lines 70-75: The authors should clarify the origin of the carbamazepine cocrystals. Where they already described in the literature?

Answer

All three studied cocrystals were previously described in the literature. References to works describing methods of preparation and structures of these cocrystals are given in the text of the manuscript. For the [CBZ+BZA] (1:1) cocrystal, single crystals were grown for the first time, and the crystal structure was solved.

Reviewer_3

2 - Line 157. Please insert the column brand. Also, add a representative chromatogram of each cocrystal analyzed in Supplementary section.

Answer

The column brand was inserted. The chromatograms of each cocrystal were added in Supplementary.

Round 2

Reviewer 1 Report

Thank you for considering my comments and suggestions. 

I still believe that the work requires some editorial corrections. In particular, the entries of the N-H… O type (eg line 279) should be corrected, especially since the rest of the work was written correctly, ie N-H ··· O.

Since the compounds CBZ + BZA] (1:1) and [CBZ + INAM] (1:1) form II are isostructural, this should be clearly stated.

Author Response

Reviewer_1

Thank you for considering my comments and suggestions. 

Reviewer_1

I still believe that the work requires some editorial corrections. In particular, the entries of the N-H… O type (eg line 279) should be corrected, especially since the rest of the work was written correctly, ie N-H ··· O.

Answer

The editorial corrections have been introduced.

Reviewer_1

Since the compounds CBZ + BZA] (1:1) and [CBZ + INAM] (1:1) form II are isostructural, this should be clearly stated.

Answer

The text has been revised and explanation has been added.
